# Maternal Genotype and Dietary Vitamin A Modify Aortic Arch Phenotypes in a Mouse Model of 22q11DS

**DOI:** 10.3390/ijms262110595

**Published:** 2025-10-30

**Authors:** Emilia Amengual-Cladera, Maria Victòria Llull-Alberti, Marc Ventayol-Guirado, Juan Antonio Jimenez-Barcelo, Jairo Enrique Rocha, Josep Muncunill, Jessica Hernandez-Rodriguez, Daniela Medina-Chávez, Elionor Lynton-Pons, Paula Sureda-Horrach, Victor Jose Asensio, Laura Ruiz-Guerra, Albert Tubau, Miguel Juan-Clar, Marchesa Bilio, Bernice Morrow, Cristófol Vives-Bauzà, Gabriella Lania, Elizabeth Illingworth, Antonio Baldini, Alexander Damian Heine-Suñer

**Affiliations:** 1Genomics of Health Research Group, Institute of Health Sciences Research of the Balearic Islands (IdISBa), 07120 Palma, Balearic Islands, Spain; mariavictoria.llull@idisba.es (M.V.L.-A.); marc.ventayol@idisba.es (M.V.-G.); juanantonio.jimenez@idisba.es (J.A.J.-B.); elionorlp@gmail.com (E.L.-P.);; 2Grup de Biologia Computacional i Bioinformàtica (BIOCOM), Institute of Health Sciences Research of the Balearic Islands (IdISBa), 07120 Palma, Balearic Islands, Spain; 3Grup de Biologia Computacional i Bioinformàtica (BIOCOM), Universitat de les Illes Balears, 07122 Palma, Balearic Islands, Spain; 4Genomics and Bioinformatics Platform, Balearic Islands Health Research Institute (IdISBa), University Hospital Son Espases, 07120 Palma, Balearic Islands, Spain; 5Molecular Diagnostics and Clinical Genetics Unit (UDMGC), Hospital Universitari Son Espases, Carretera de Valldemossa 79, 07120 Palma, Balearic Islands, Spain; 6Grupo Neurobiología, University of Balearic Islands (UIB), 07122 Palma, Balearic Islands, Spaincristofol.vives@uib.cat (C.V.-B.); 7Servicio de Ginecología y Obstetricia, Hospital Universitario Son Llàtzer, 07198 Palma, Balearic Islands, Spain; 8Institute of Genetics and Biophysics Adriano Buzzati-Traverso, 80131 Naples, Italy; 9Department of Genetics, Albert Einstein College of Medicine, Bronx, NY 10461, USA; 10Grup de Neurobiologia, Institute of Health Sciences Research of the Balearic Islands (IdISBa), 07120 Palma, Balearic Islands, Spain; 11Department of Chemistry and Biology, University of Salerno, 84084 Fisciano, Italy; 12Department of Molecular Medicine and Medical Biotechnology, University Federico II, 80138 Naples, Italy

**Keywords:** 22q11.2 deletion syndrome, congenital heart defects, vitamin A

## Abstract

Congenital heart defects (CHDs) occur in 50–75% of patients with 22q11.2 deletion syndrome (22q11.2DS), ranging from mild to severe manifestations. The genetic and environmental factors contributing to variable CHD phenotypes in 22q11.2DS are largely unknown. In this study, we used a mouse model of 22q11.2DS, termed Df1/+, to evaluate the effect of maternal vitamin A (VitA) dietary imbalance (supplementation or deficiency) on the incidence of aortic arch defects (AADs), which is a common type of CHD observed in both 22q11.2DS patients and Df1/+ mouse embryos. While most groups showed a previously observed 30% AAD incidence, two groups exhibited significantly higher rates: (1) Df1/+ embryos from WT mothers on a VitA-Supl diet (51% AADs) and (2) Df1/+ embryos from Df1/+ mothers on a VitA-Def diet (45% AADs). Thus, a low or high maternal VitA diet can increase the frequency of AADs in embryos depending on the maternal genotype. Transcriptomic analysis of the hearts of these high-risk embryos at embryonic day (E)18.5 revealed downregulation of key genes (*Hdac3*, *Ptgds*, *Sirt5*, *Pfkm,* and *Lclat1*) associated with energy metabolism pathways, such as oxidative phosphorylation and glycolysis, suggesting impaired cardiac recovery mechanisms. In conclusion, our findings demonstrate that altered VitA exposure can exacerbate AAD incidence in a maternal-genotype-dependent manner, highlighting the complex interplay between embryonic and maternal genetic background and environmental factors in CHDs associated with 22q11.2DS.

## 1. Introduction

Congenital heart defects (CHD) are structural heart malformations that arise during embryogenesis. They are the leading cause of neonatal morbidity and mortality, with an incidence of nearly 1–2% in newborns, and are responsible for up to 30% of lethality during infancy unless repaired by surgery [1,2]. In Europe, these cardiac malformations have an impact of about 7/1000 live births, although the impact would be even greater if one considers foetuses that fail to reach term [1,2,3]. To date, many of the genetic and molecular bases of CHDs remain undiscovered, with genetic factors accounting for, at most, 25% of cases [4]. Therefore, studies using mouse models are essential to deepen our understanding of the underlying causes of CHDs and to advance diagnostic and therapeutic strategies for this disorder.

CHDs can be classified as syndromic or non-syndromic depending on whether the patient has additional neurodevelopmental abnormalities or other organ malformations [5,6]. Most non-syndromic CHDs occur sporadically, with a very scarce number of families having a clear monogenic inheritance. This makes the identification of the genetic cause very difficult using a classical genetic approach [7]. Syndromic CHDs can result from chromosomal aneuploidies like Down syndrome (trisomy 21) and Turner syndrome (monosomy X) or from pathogenic copy number variations (CNVs) such as Williams syndrome (7q11.23 deletion) or 22q11.2 deletion syndrome (22q11DS). Besides these examples, identifying the genetic causes through classical genetic approaches is challenging, leading to the current understanding that most CHDs have a multifactorial origin, involving both common and rare genetic variants influenced by adverse environmental factors [4,5,6,8]. In this context, studies of rare disorders with well-characterized etiologies can serve as model systems to discover and understand the genetic and environmental risk factors for CHDs.

22q11.2DS is the most common human chromosomal deletion syndrome, affecting approximately 1 in 4000 individuals and 1 in 1000 fetuses [9]. The 22q11.2 region, flanked by low-copy repeats, is susceptible to misalignment during meiosis, leading to large interstitial genomic deletions. Around 90% of individuals with 22q11.2DS carry a 3 Mb deletion, while ~8% have a nested 1.5 or 2 Mb deletion [10,11,12]. The frequency and types of CHDs observed in individuals with nested 1.5 or 2 Mb deletions are similar to those with the typical 3 Mb deletion, indicating that genes in the nested 1.5 Mb region are key contributors to the CHD phenotype. Among the genes in this region, TBX1, encoding a T-box transcription factor, is considered a major driver of the syndrome [13,14,15], while genes located in the other half of the region, such as CRKL, further modulate the CHD phenotype [16,17]. In any case, 60–80% of individuals sharing the minimum deletion of 1.5 Mb show a CHD [18]. Additionally, cardiac malformations range from mild to severe and include conotruncal heart defects such as AADs, tetralogy of Fallot, and persistent truncus arteriosus.

Research in the field suggests the existence of genetic modifiers of CHD in 22q11.2DS, but these explain only a subset of the observed variability. There was one report of functional single-nucleotide polymorphisms in the promoter of the VEGFA gene at 6p12, found to increase the risk of CHDs in both 22q11.2DS patients and in VEGF-deficient mouse models [19,20]. Additionally, common variants in the remaining allele of 22q11.2 [21] and rare single-nucleotide variants in chromatin regulatory genes across the genome [22,23] explain some of the possible genetic variation, though only in a subset of subjects. Thus, major gaps remain in our knowledge of the full range of genetic modifiers across the genome that contribute to these phenotypic differences. Furthermore, environmental factors likely play a significant role in modifying the overall cardiac phenotype in 22q11.2DS patients. To address this, one goal of this work is to investigate the impact of maternal genotypes and environmental factors on CHD risk in 22q11.2DS using a mouse model carrying the 1.5 Mb 22q11.2 deletion, referred to as Df1/+ [24].

The human 22q11.2 region and genes within it are largely conserved on mouse chromosome 16 [25,26]. This evolutionary conservation, despite some rearrangements in gene order, enables precise modelling of the nested 1.5 Mb deletion in 22q11.2DS in mice through gene targeting approaches [13,24]. The Df1/+ mouse, developed by Baldini’s group in 1999, carries a hemizygous deletion of the equivalent 1.5 Mb region on mouse chromosome 16, spanning from the *Ess2* gene to the *Ufd1l* gene. This region contains about 26 coding genes, including *Tbx1*, and 13 non-coding genes or predicted genes. Approximately 30% of Df1/+ embryos exhibit CHDs at birth, with AADs being the predominant type. Under normal conditions, the right fourth pharyngeal arch artery (PAA) gives rise to the right subclavian artery, while the left fourth PAA contributes to a critical portion of the aortic arch [13,24]. In Df1/+ embryos, however, these developmental processes are frequently disrupted, leading to abnormal remodeling of the aortic arches and the consequent CHD characteristic of this model. Such disruptions in Df1/+ embryos or in patients with 22q11.2DS can result in vascular anomalies such as a retroesophageal or aberrant right subclavian artery (reRSA/ARSA) or an interrupted aortic arch type B [24]. Notably, at embryonic day (E) 10.5, 100% of Df1/+ mice exhibited hypoplasia or aplasia of one or both fourth PAAs. However, only 30% presented with an aortic arch defect (AAD) at birth, suggesting partial recovery of the affected arteries during later stages of embryogenesis, although to date, the mechanisms and processes involved are unknown [27]. Thus, altogether, the above makes the Df1/+ mouse a valuable model for studying the impact of environmental factors as modifiers of the cardiac phenotype.

One such environmental factor is retinoic acid (RA), the active form of vitamin A (VitA). Alterations in RA metabolism have been linked to aortic arch defects (AADs) similar to those observed in 22q11.2 deletion syndrome (22q11.2DS) [28,29]. In mammalian embryos, retinol is first converted to retinal by retinol dehydrogenases (RDH) and subsequently to RA by retinaldehyde dehydrogenases (RALDH). Mouse mutants lacking RALDH2 exhibit key features of 22q11.2DS, including aortic arch anomalies [30,31].

Moreover, emerging evidence suggests that maternal diet, particularly VitA intake during pregnancy, may influence the phenotype in 22q11.2DS [32]. Several studies have demonstrated strong interactions between nutritional and genetic factors in the development of CHD [33,34,35,36]. However, little is known about how maternal genotype interacts with dietary interventions in the context of genetic syndromes such as 22q11.2DS.

In this study, we aimed to investigate how non-teratogenic imbalances in maternal dietary VitA during pregnancy, in combination with both maternal and embryonic 22q11.2DS genotypes, affect the incidence of AADs using the Df1/+ mouse model.

## 2. Results

### 2.1. Experimental Setup and Groups

Three main experimental groups were defined based on the maternal dietary dose of vitamin A: VitA supplementation (VitA-Sup), VitA control (VitA-Cont), or VitA deficiency (VitA-Def) (Figure 1). WT (wild-type) and Df1/+ (22q11.2DS mouse model) mice were mated with males with the opposite genotype. Female mice were fed with one of those diets for 10 weeks prior to mating and throughout pregnancy (for details, see the Materials and Methods). At E18.5, embryos were phenotyped and genotyped. Embryos were classified according to genotype and cardiac phenotype as follows: WT or Df1/+ and Normal (no aortic arch defect) or AAD (aortic arch defect).

### 2.2. Litter Size Is Modulated by Maternal Genotype and Dietary Vitamin A Dosage

#### 2.2.1. Vitamin A in the Diet Determines Retinol Levels in the Blood

To assess the impact of dietary VitA dosage on physiological retinol levels, we measured retinol concentrations in the maternal serum. Mothers on the VitA-Sup diet showed a statistically significant twofold increase in serum retinol levels compared to those on the control diet, while VitA-Def mothers had levels reduced to half that of the control group (Figure 2A). Overall, serum retinol levels in VitA-Sup mice were 3.6 times higher than those of VitA-Def mice. This pattern was independent of the mother’s genotype (WT or Df1/+). Moreover, no differences in food intake were observed between groups, indicating that differences in serum retinol levels were due to variations in dietary VitA dosage.

#### 2.2.2. Litter Size

We phenotyped a total of 432 embryos, 256 from WT mothers and 176 from Df1/+ mothers, that came from 66 litters, of which 37 were from WT mothers and 29 were from Df1/+ mothers (Table 1). We found that litter sizes from Df1/+ mothers were, on average, slightly non-significantly lower than those from WT mothers (6.0 embryos/litter Df1/+ mothers vs. 6.9 WT mothers). When embryos were harvested, we also determined the number of reabsorptions (non-viable embryos) in each litter, and if we take into account reabsorptions per litter (7.1 embryos + reabsorptions Df1/+ mothers vs. 7.7 embryos + reabsorptions WT mothers), this does not change greatly. In summary, excluding sampling errors, although fertility seems slightly lower in Df1/+ mothers when compared to WT mothers, this difference is non-significant.

#### 2.2.3. Df1/+ Embryos vs. WT Embryos

Overall, we found no deviation from the expected 50:50 ratio of WT and Df1/+ embryos (216 Df1/+ vs. 216 WT) (Figure 2B). This distribution remained consistent regardless of the mother’s genotype (Df1/+ mothers: 85 Df1/+ vs. 89 WT/+ embryos; WT mothers: 129 Df1/+ vs. 125 WT embryos). However, when stratifying by diet, we observed that VitA-Sup had a trend towards a smaller average litter size compared to the VitA-Def and Control diet groups, suggesting that supplementation might negatively affect embryo viability (Figure 2B).

Analysis of reabsorptions (non-viable embryos) revealed that altered diets (VitA-Sup and VitA-Def) result in significantly higher reabsorption rates compared to the control diet (VitA-Def: 19.4%; VitA-Sup: 25.7% vs. VitA-Control: 7.5% of total embryos). Notably, this increase was only observed in embryos from Df1/+ mothers, not from WT mothers (Figure 2C). Accordingly, the lower average number of embryos from Df1 mothers could be explained by a significant increase in reabsorptions in these groups, indicating that embryos from WT mothers exhibit greater viability than those from Df1 mothers (Figure 2C).

### 2.3. Frequency of Aortic Arch Defects (AADs)

Approximately 30% of heterozygous Df1/+ mice have been reported to exhibit aortic arch defects at birth [24]. These abnormalities primarily include IAA-B, reRSA/ARSA, and right aortic arch. To confirm that our mice did not present with additional cardiac malformations beyond these types, we conducted histological examinations of the hearts from 55 embryos using H&E staining. In this subsample, we observed 42 normal phenotypes and 13 AADs but found no septal or other internal structural heart abnormalities. Based on these findings, we proceeded to phenotype the remaining hearts and great vessels by visual inspection alone.

In total, phenotypic characterization identified 76 embryos with AADs (76/432; ~18%). Of these, 69 embryos carried the Df1/+ genotype (69/216; ~32%), while 7 were wild type (7/216; ~3%). The most common aortic arch defects detected were reRSA/ARSA (n = 56 embryos) and IAA-B (n = 17 embryos) (Figure 2D). In addition, we found 4 embryos with combinations of two cardiac malformations, 3 with both ARSA and IAA-B, and 1 with ARSA and a right aortic arch (Figure 2D).

### 2.4. AAD Penetrance in Df1/+ Embryos Is Modulated by Maternal Genotype and Dietary VitA Dosage

In our study, all experimental groups of Df1/+ embryos, except two, showed approximately 30% incidence of AADs at E18.5, consistent with the findings by Lindsay and Baldini [27]. However, two experimental groups showed a markedly higher incidence of AADs, as observed in Figure 2E. Notably, Df1/+ embryos from WT mothers fed a VitA-Sup diet exhibited a 51% incidence of AADs, representing a 21% increase compared to the control diet. In addition, Df1/+ embryos from Df1/+ mothers fed a VitA-Def diet displayed a 45% incidence of AADs, 15% higher than in the control diet group. Similar trends were observed in WT embryos (Figure 2F): in WT mothers, VitA-Sup increases the incidence of AADs two-fold compared to VitA-Def, while in Df1/+ mothers, the incidence of AADs in Df1/+ embryos was two-fold higher in the VitA-Def diet group than in the VitA-Sup group.

The two groups were classified as ‘High-Risk Group 1′ (HRG1) and ‘High-Risk Group 2′ (HRG2). Specifically, HRG1 includes Df1/+ embryos from WT mothers fed a VitA-Sup diet, while HRG2 comprises embryos from Df1/+ mothers on a VitA-Def diet. In both groups, embryos were subdivided into AAD-affected and phenotypically normal, with approximately half of the embryos falling into each category (HRG1-AAD/Normal: 51%/49%; HRG2-AAD/Normal: 45%/55%) (Figure 2E).

These phenotypic findings highlight a striking interaction between maternal genotype and dietary VitA status, suggesting that the maternal background might modulate embryonic susceptibility to either VitA excess or deficiency.

### 2.5. Differential Gene Expression Analysis

To determine whether the cardiac gene expression of Df1/+ embryos is influenced by maternal diet and maternal genotype, we performed differential gene expression analysis using TAC software to identify differentially expressed genes (DEGs) and gene set enrichment analysis (GSEA) to determine the most significantly affected biological processes (BPs).

#### 2.5.1. Gene Expresson and Pathway Analysis

Following this analysis, HRG1-AAD and HRG2-AAD Df1/+ embryos, with an AAD, were independently compared to their Df1/+ counterparts, serving as controls that differed in only one variable, either maternal diet or maternal genotype. Through this comparative analysis, our aim was to characterize which specific transcriptional differences are associated with each condition. For that reason, we classified gene expression differences into two categories: BPs linked to the mother’s diet (Figure 3(C1,C2)) and BPs linked to the mother’s genotype (Figure 3(C3,C4)).

#### 2.5.2. Changes in Embryonic Cardiac Gene Expression Linked to the Mother’s Diet

To assess whether and how maternal VitA doses influenced HRG1-AAD and HRG2-AAD embryos, we performed the following comparisons:Comparison 1 (C1): Cardiac gene expression from Df1/+ embryos with AADs from WT mothers fed a VitA-Sup diet (HRG1-AAD) versus Df1/+ embryos with AADs from WT mothers fed a VitA-Def diet (non-HRG1) (Figure 3(C1)).Comparison 2 (C2): Cardiac gene expression from Df1/+ embryos with an AAD from Df1/+ mothers fed a VitA-Def diet (HRG2-AAD) versus cardiac gene expression from Df1/+ embryos with an AAD from Df1/+ mothers fed a VitA-Sup diet (non-HRG2) (Figure 3(C2)).

The top 20 most significantly altered BPs of C1 were all downregulated in HRG1-AAD embryos compared to their counterparts. In contrast, the C2 comparison showed a combination of both upregulated (60%) and downregulated BPs (40%). Notably, both HRG-AAD groups in the C1 and C2 comparisons shared several downregulated BPs, including those related to heart development (GO:0048738, GO:0003007, and GO:0007512), cytoskeletal assembly (GO:0031032 and GO:1903115), and energy metabolism, such as the generation of precursor metabolites and energy (GO:006091) and NADH metabolic process (GO:0006120). These expression differences were observed among embryos with an AAD from the same maternal genotype, differing only in maternal diet, which suggests a direct relationship with this factor.

#### 2.5.3. Changes in Embryonic Cardiac Gene Expression Linked to the Mother’s Genotype

To assess whether and how maternal genotype influences gene expression related to the cardiac phenotype in Df1/+ embryos under identical dietary conditions, we performed the following comparisons:Comparison 3 (C3): Cardiac gene expression from Df1/+ embryos with AADs from WT mothers fed a VitA-Sup diet (HRG1-AAD) versus cardiac gene expression from Df1/+ embryos with an AAD from Df1/+ mothers fed a VitA-Sup diet (non-HRG2) (Figure 3(C3)).Comparison 4 (C4): Cardiac gene expression of Df1/+ embryos with an AAD born from Df1/+ mothers fed a VitA-Def diet (HRG2-AAD) versus cardiac gene expression of Df1/+ embryos with an AAD born from WT mothers fed a VitA-Def diet (non-HRG1) (Figure 3(C4)).

Enrichment analysis revealed that, in C3, 90% of the top 20 significantly affected BPs were upregulated compared to the non-HRG. In contrast, in C4, all of the top 20 BPs were downregulated (Figure 3(C3,C4)). Interestingly, despite these differences, both HRGs (HRG1-AAD and HRG2-AAD) exhibited downregulation of energy metabolic processes such as mitochondrial respiratory chain complex assembly (GO:0033108) and energy derivation through oxidation of organic compounds (GO:0015980) compared to their pseudo-controls (non-HRG and non-HRG). These expression differences are between embryos all with an AAD from the same maternal diet and that only differ in the maternal genotype, which suggests a direct relation to this last factor.

#### 2.5.4. Changes in Embryonic Cardiac Gene Expression Are Linked to the Presence of an AAD

To identify which of the transcriptomic differences observed is associated with the presence of an AAD compared to WT in HRG embryos, we compared HRG1-AAD and HRG2-AAD with HRG1-Normal and HRG2-Normal, respectively (Figure 3(C5,C6)).

GSEA revealed that HGR1-AAD and HGR2-AAD exhibited significant widespread downregulation of BPs compared to HRG embryos with a normal cardiac phenotype. Notably, the most significantly downregulated BP shared by HRG1-AAD and HRG2-AAD embryos was aerobic respiration (GO:0009060), which exhibited both the highest normalized enrichment score (NES) and the greatest gene ratio. Additionally, other downregulated BPs related to energy metabolism were identified, with ATP metabolic process (GO:0046034) downregulated in one group and NADH dehydrogenase complex assembly (GO:0010257) in the other. The overlap of downregulated BPs with similar functions in embryos with an AAD from both comparisons suggests that these processes may result from the presence of an AAD or be involved in the pathogenesis of these defects.

### 2.6. Gene Expression and Candidate Genes

As stated in the previous sections, GSEA results indicated that, although HRG1-AAD and HRG2-AAD embryo groups originate from mothers with different genotypes and diets, there were commonly affected BPs across both groups. This suggested that there were genes involved in similar pathways that were consistently dysregulated. These common genes that are dysregulated may serve as molecular indicators of pathological mechanisms initiated at earlier developmental stages.

To identify common DEGs between HRG1-AAD and HRG2-AAD, we used TAC software and created a Venn diagram of shared DEGs between each of the previously described comparisons (C1–C6) (Figure 4). This analysis identified 78 shared DEGs between comparisons C1 and C2 (gene expression linked to different VitA dosage in the maternal diet) (Figure 4A), 117 shared DEGs between comparisons C3 and C4 (gene expression linked to different maternal genotype) (Figure 4B), and 14 shared DEGs between comparisons C5 and C6 (gene expression linked to the presence of an AAD) (Figure 4C). The above-described DEGs add up to a total of 210 candidate genes, as 6 of them are found in more than one of the comparison groups (see Appendix A).

Additionally, we conducted a scatterplot analysis to illustrate the direction of the expression changes for shared genes across comparisons (Figure 4). These plots show that shared genes in HRG1-AAD and HRG2-AAD may exhibit either concordant expression patterns (e.g., upregulation–upregulation or downregulation–downregulation) or discordant patterns (upregulation–downregulation or vice versa) when compared to non-HRG1 and non-HRG2 and HRG with a normal phenotype across all comparisons (C1–C6) (Figure 4A–C).

The Venn diagram analysis revealed that among the 210 shared genes, only *Hdac3* and *Ptgds* appeared in all comparisons (C1–C6) (Figure 4). Both genes were consistently downregulated across all comparisons, showing reduced expression in Df1/+ embryo groups with higher AAD incidence compared to those with a normal phenotype (C1–C2). Similarly, *Hdac3* and *Ptgds* also appeared to be downregulated when comparing Df1/+ embryos exhibiting AAD phenotypes, when comparing them from mothers with different diets or maternal genotypes (C3 and C6) (see Figure 4).

In addition, other common DEGs were identified in maternal diet or maternal genotype (C1-C4) pairwise comparisons. In these, we observed differential gene expression of *Sirt5*, *Pfkm,* and *Lclat1* in both the diet-based and maternal genotype-based comparisons (Figure 5). Similar to *Hdac3* and *Ptgds*, all of these genes displayed a consistently downregulated expression profile across all comparisons (see Figure 4).

## 3. Discussion

Our study reveals that both maternal genotype and maternal dietary VitA doses can significantly influence phenotypic variability in 22q11.2DS embryos. This represents a novel insight, as most prior studies have focused primarily on the embryonic genotype and VitA levels. By identifying two high-risk groups with increased AAD penetrance, our findings suggest that maternal factors can modulate the severity of CHD.

Cardiac remodeling of the pharyngeal arches during fetal development is crucial for the proper formation of the major blood vessels emerging from the heart. These arches originate from the aortic sac, which arises from the primary heart tube, and form sequentially through a complex remodeling process [37]. In mice, the third and fourth pharyngeal arches—precursors of the aortic arch—begin to form at E9.5 and undergo remodeling between E10.5 and E11.5 [37,38,39]. This process involves contributions from both second heart field (SHF) cells and cardiac neural crest cells (cNCCs), which give rise to the smooth muscle cells (SMCs) that populate the walls of the aortic arch [40].

It is important to emphasize that the development of the heart and the aortic arch is closely interconnected. Both structures originate, at least in part, from SHF-derived cells and are shaped by complex genetic and cellular interactions involving additional populations such as first heart field cells, cNCCs, and cells from the proepicardial organ. As a result, these structures share embryonic origins and are governed by overlapping molecular and genetic pathways. Consequently, alterations in gene expression within the heart can adversely affect the remodeling of the pharyngeal arches [41,42].

Mouse models of 22q11.2DS, such as Df1, exhibit hypoplasia of the fourth PAA embryos by E10.5, which can progress into an AAD at later stages. However, not all embryos progress to AAD, suggesting that compensatory mechanisms may enable some phenotypic recovery. Initial observations in the Df1 mouse model demonstrated that 100% of embryos showed PAA defects at E10.5 but only 30% progressed to AADs by birth [14]. These observations were subsequently confirmed in other studies [43,44,45,46,47,48,49].

In a recent meta-analysis by Anderson and Bamforth [50] based on 10 independent studies using either Tbx1 knockout or Df1 22q11.2DS mouse models, PAA defects were observed in an average of 81.7% of embryos at E10.5, decreasing to 28.7% by E14.5. Similar findings have been noted in the Lgdel mouse model, which carries a larger deletion than Df1 [32]. Additionally, other studies suggest that the genetic background of embryos may affect the recovery process, with C57BL/6 mice showing a higher AAD penetrance at term than 129SvEv mice, though they did not find any difference in the penetrance of the fourth PAA phenotype at E10.5 [51]. Therefore, the Df1 model provides a stable background to test specific environmental factors, such as dietary VitA, on the development of CHD, making it a suitable system for investigating therapeutic interventions and the mechanisms underlying phenotypic recovery.

Retinoic acid (RA), a biologically active metabolite of VitA, plays a crucial role in embryonic cardiac patterning by regulating the development and differentiation of cardiac cells during early embryogenesis [31,32,52]. Deficiency of RA has been shown to accelerate recovery from arterial growth delay in Tbx1+/- mice [31]. Furthermore, it has also been shown that reduced VitA in the maternal diet produces embryos at E10.5 with fewer PAA abnormalities than those with higher doses in an Lgdel 22q11DS mouse model (4 vs. 16 IU VitA) [32]. These observations support the hypothesis that reduced VitA intake during pregnancy could help minimize congenital heart defects, particularly AADs, in offspring affected by 22q11.2DS [28,29,30,31]. On the contrary, increased levels of RA derived from VitA supplementation would be predicted to exacerbate the 22q11.2DS cardiac phenotype in the same context [53].

Our findings reveal a more complex scenario, in which the interplay between dietary VitA levels and both the embryonic and maternal 22q11.2DS genotypes significantly influences the cardiac outcome. We found that VitA supplementation was associated with a higher incidence of AADs only in embryos from WT mothers, whereas under deficiency, increased AAD numbers were observed exclusively in embryos from Df1/+ mothers (Figure 2E). We designated these two groups as High-Risk Groups (HRG1 and HRG2, respectively), as they seem to reflect a greater predisposition to developing AADs on embryonic day E18.5 compared to other studies. These results underscore a complex interplay by which both excess and deficiency in VitA can increase AAD incidence, depending on maternal genotype.

We find that blood retinol levels and embryo genotypic ratios do not explain the observed HRG effect. Consequently, there is no straightforward explanation for the existence of the two distinct high-risk groups, one of them affecting WT mothers and the other affecting Df1/+ mothers.

Our transcriptomic analyses were performed at E18.5, when the aortic arch is already formed. Even so, it revealed expression differences between embryos with AADs from HRG (higher than 30% of AAD incidence) and embryos with AADs that are not classified as HRG (incidence of an AAD around 30%). This may suggest that there are pathogenic mechanisms at play that lead to the AAD at E18.5 that may be different between Df1/+ embryos from HRG and those from other groups, with the expected AAD incidence. Although at E18.5 most of the developmental processes are completed or being completed, the transcriptomic differences found may still allow us to infer some of these genetic mechanisms at play in the HRG groups.

Moreover, GSEA revealed a consistent downregulation of energy metabolism in comparisons addressing dietary conditions and maternal genotype between HRG and non-HRG embryos with an AAD (Figure 3). These similarly altered BPs at E18.5 that are not a direct consequence of the aortic arch malformation itself, as both comparison groups carry an AAD, suggest that there are transcriptomic similarities between HRG embryos. This reflects shared disruptions in biological pathways that serve as molecular indicators of cardiac defects present in HRG1 and HRG2 that were present during earlier developmental stages. Furthermore, we identified five common genes that showed lower expression levels in both groups, HRG1-AAD and HRG2-AAD (Figure 4): Sirtuin 5 (Sirt5), Phosphofructokinase 1 (Pfkm), Acyl-coenzyme A: lysocardiolipin acyltransferase-1 (Alcat1/Lclat1), Histone deacetylase 3 (Hdac3), and Prostaglandin D2 Synthase (Ptgds). Notably, these five genes are associated with common similar significantly altered biological processes that were also present in the GSEA analysis: energy metabolism, histone modification, and vascular structure formation.

Mitochondria play a crucial role in heart development, as the heart is a highly energy-demanding organ that relies on mitochondrial function for proper growth, differentiation, and maturation [54]. During early heart development, this energy is primarily obtained through glycolysis, which occurs within the cytosol [55,56]. Three of the common genes (Sirt5, Pfkm, and Lclat1) are directly involved in maintaining oxidative metabolism in the mitochondria to support cell survival. First, Sirt5 encodes an NAD+-dependent deacylase that responds to oxidative stress by deacetylating proteins involved in oxidative cellular metabolism as an adaptive mechanism [57]. Second, Pfkm encodes a rate-limiting enzyme in the glycolytic pathway, the primary energy source during heart development [56]. Third, Lclat1 encodes an acyltransferase that promotes the turnover of cardiolipins in the mitochondria, ensuring the integrity of the inner mitochondrial membrane, which is essential for the proper function of the respiratory chain [58]. Disruptions in the expression of Sirt5, Pfkm, and Lclat1 have been associated with mitochondrial dysfunction and also the onset of congenital heart disease, highlighting their importance in heart development [56,57,58].

The remaining two genes, Hdac3 and Ptgds, were identified in all three comparisons, including differences between Df1 embryos with and without an AAD. This may indicate that lower expression of these two genes could be triggered by all three intervening factors: maternal diet, maternal genotype, and embryos’ phenotype. Ptgds encodes an enzyme responsible for catalyzing the production of PGD2, a prostaglandin with diverse roles in inflammation, immune response, sleep regulation, and cardiovascular function. Notably, PGD2 contributes to cardio protection and vascular homeostasis, influencing blood pressure and cardiovascular stability [59].

Of particular interest, Hdac3 is a gene closely associated with heart development, RA signaling, and mitochondrial metabolism. It encodes histone deacetylase 3, a class I HDAC that functions as a key epigenetic regulator, modulating gene expression and cellular metabolism through histone deacetylation, chromatin condensation, and transcriptional repression [60]. HDAC3 plays a critical role in RA signaling by repressing gene expression in the absence of RA and permitting activation in its presence [61]. This dual regulatory function suggests that HDAC3 may exacerbate developmental defects in the context of Tbx1 haploinsufficiency, particularly affecting cardiac morphogenesis. Supporting this idea, recent studies have shown that mutations in chromatin-modifying genes increase the risk of CHDs in individuals with 22q11.2DS 22. In addition, HDAC3 is known to regulate smooth muscle differentiation, and its loss in neural crest cells leads to severe cardiovascular malformations, including IAA-B, aortic arch hypoplasia, double outlet right ventricle, and ventricular septal defects [62]. HDAC3 also modulates cardiac metabolism by regulating genes involved in oxidative phosphorylation and mitochondrial bioenergetics [63]. Moreover, Hdac3-null mice show heightened sensitivity to dietary perturbations, such as high-fat diets, which can lead to lethality due to impaired mitochondrial gene expression in the myocardium [64]. Altogether, these findings position HDAC3 as a central epigenetic mediator connecting RA signaling, mitochondrial function, and cardiac development, pathways that we have identified as compromised in HRG1 and HRG2. However, further research is needed to elucidate the precise roles of HDAC3 and other newly implicated genes in the pathogenesis of AADs.

Overall, our data suggests that both maternal diet and genotype can influence the phenotype of mouse embryos carrying the equivalent deletion of DiGeorge Syndrome. Also, we found evidence that these factors may interfere with compensatory mechanisms during cardiac remodeling, potentially leading to transcriptomic alterations detectable at E18.5.

## 4. Conclusions

In conclusion, our results reveal that both VitA deficiency and supplementation can increase AAD penetrance but only in the context of specific maternal genotypes. This adds a new layer to our understanding of CHD in 22q11.2DS and highlights the importance of maternal influences. Given that pregnancies in women with 22q11.2DS are common, and their children may present more severe phenotypes, these findings could inform future dietary guidelines and therapeutic strategies.

For future research, it will be necessary to consider additional developmental stages. Nonetheless, it is important to emphasize that this study represents an initial approach to understanding the complex interplay between maternal VitA intake and genetic variants such as the 22q11.2 deletion. Broadening this scope will contribute to a more comprehensive understanding of the field and provide stronger support for emerging researchers.

## 5. Limitations of the Research

One of the main limitations of our study is that the congenital heart defects observed in the phenotypic analysis were exclusively aortic arch anomalies, whereas the transcriptomic analysis was performed using heart tissue, including the ascending aorta. In addition, the cardiac phenotype was only evaluated by visual assessment to detect aortic arch defects. Another limitation is the use of samples collected at E18.5, which is an embryonic stage where the aortic arch is fully formed and thus, transcriptome changes do not directly interrogate remodeling processes that may be at play.

## 6. Materials and Methods

### 6.1. Mice and Diets

All animal procedures used in this study were performed in accordance with the general guidelines approved by our institutional animal welfare ethics committee and EU regulations (CEEA 13/03/14; 2010/63/UE). Df1/+ female mice bred into a C57Bl6 background and wild-type (WT) C57BL/6 male mice were obtained from the European Mouse Mutant Archive (EMMA)—Infrafrontier (Salerno, Italy). All animals were kept under controlled conditions (22 °C and 65 ± 3% humidity) with a 12 h light–darkness cycle with free access to food and water. We randomly selected 4-week-old WT females and assigned them to 3 experimental groups (Figure 1).

Vitamin A dosage in deficient, supplemented, and control diets was selected to avoid teratogenicity, as published by Niederreither [65]. Each group was fed one of three different diets for 10 weeks prior to mating and throughout pregnancy. The control group was fed a standard diet containing 20 IU vitamin A/Kg (VitA-Cont, TD 91280, Harlan, Kentucky, USA), the supplemented group received a VitA-enriched diet containing 10 times the recommended dietary amount of VitA (VitA-Sup, 200 IU vitamin A/Kg diet, TD 110146, Harlan), and the deficient group was fed a VitA-free diet with no VitA (VitA-Def, 0 IU vitamin A/Kg diet, TD 86143, Harlan) (Figure 5). In parallel, 4-week-old Df1/+ females were also divided into 3 experimental groups and fed the same experimental diets (VitA-Cont, VitA-Sup, and VitA-Def).

Food intake was assessed daily by weight difference for 3 weeks to ensure that no discrepancies in diet consumption existed between groups. After treatment, both WT and Df1/+ females were mated overnight with Df1/+ or WT males, respectively, who had been fed a standard rodent diet 5LF2 (EURodent diet 14%, LabDiet, St. Louis, MO, USA) from weaning. The following morning, females were checked for the presence of a vaginal plug, and if so, that morning was designated as embryonic day (E) 0.5. Pregnant females were kept under the same dietary treatment until they were sacrificed on E18.5 by means of CO_2_ exposure. Maternal blood was collected by means of cardiac puncture after a 12 h fasting period. Serum was obtained via centrifugation at 1200× *g* for 10 min and stored at −80 °C until analysis. Total maternal serum VitA (retinol) levels were measured using an ultra-resolution liquid chromatography/photodiode detector, with the procedure performed by an external analysis laboratory (Echevarne, Barcelona, Spain). Embryos were collected, genotyped to distinguish WT from Df1/+ embryos, and phenotyped to identify cardiovascular malformations. Reabsorption sites were also recorded.

As a result of the genotyping and phenotyping procedures, embryos from each experimental diet group (VitA-Control, VitA-Sup, or VitA-Def) were classified into three categories per WT mother: WT embryos with a normal phenotype, Df1/+ embryos with a normal cardiac phenotype, and Df1/+ embryos with AADs. This resulted in a total of 12 embryo groups from WT mothers and 12 groups from Df1/+ mothers subjected to different dietary conditions (Figure 1). All embryos are characterized by four variables: maternal genotype (WT or Df1/+), diet (VitA-Control, VitA-Sup, or VitA-Def), embryonic genotype (WT or Df1/+), and embryonic phenotype (Normal or AAD) (Figure 1). This experimental design allowed for the assessment of the effect of modified diets, the embryo’s genotype, and maternal genotype on the embryo’s AAD development.

#### 6.1.1. Embryo Genotyping and Sex Identification via PCR

Genomic DNA was isolated from embryonic tails using a high-salt extraction method. Genotyping PCR was performed according to a previously published method 23, with minor modifications. Briefly, three different primers were used to distinguish both genotypes in the same PCR reaction: UFDTAR 2R (5′-TGG-GCA-ATT-GTT-TAA-TCT-TCC-3′) and UFDWT3 (5′-CAG-AGT-TCT-GAC-TTC-TGC-ACA-CTA-A-3′) to amplify the genomic Ufd1l gene present in all mice and a combination of the aforementioned UFDTAR 2R primer with the UFDTAR 2F (5′-TCT-TTG-TCA-GCA-GTT-CCC-TTT-3′) primer to amplify the HPRT human gene inserted only in the 1.2 Mb deleted-Df1/+ mice genome. The thermal cycle conditions included 2 min at 95 °C followed by 45 cycles of 30 s at 95 °C, 30 s at 56 °C, and 3 min at 64.9 °C. A final step of 10 min at 64.9 °C to ensure the correct amplification of all of the transcripts was added before cooling at 4 °C.

The sex of the embryos was determined through amplification of the Rbm31xy gene. Primers were designed flanking an 84 bp deletion of the X-linked Rbm31x gene relative to its Y-linked gametolog Rbm31y. Based on this, a single product was amplified in females (269 bp) and two products were amplified in males (269 bp and 353 bp). The primers used were Rbmp31 Fw (5′-CAC-CTT-AAG-AAC-AAG-CCA-ATA-CA-3′) and Rbmp31 Rv (5′-GGC-TTG-TCC-TGA-AAA-CAT-TTG-G-3′). The thermal cycle conditions included 2 min at 95 °C followed by 45 cycles of 30 s at 95 °C, 30 s at 56 °C, and 3 min at 72 °C. A final step of 5 min at 72 °C to ensure the correct amplification of all of the transcripts was added before cooling at 4 °C.

#### 6.1.2. Phenotypic and Histological Analysis

The first batch of E18.5 embryos was isolated from the uterus and extra-embryonic membranes and fixed using a 24 h protocol with PBS-buffered 4% paraformaldehyde. Following fixation, embryos were kept in 70% ethanol and phenotyped by direct observation under a stereo microscope. Furthermore, 55 embryonic fixed hearts were collected, dehydrated, embedded in paraffin, cut into 10 µm thick sections, and hematoxylin-eosin (H&E) stained (Sigma Aldrich, St. Louis, MO, USA) for internal histologic examinations. We analyzed 7 WT embryos and 7 Df1/+ embryos from WT mothers fed the VitA-Sup diet, 10 WT embryos and 3 Df1/+ embryos from Df1/+ mothers fed the VitA-Def diet, and also 10 WT embryos and 18 Df1/+ embryos from WT mothers fed the VitA-Def diet. Once the histological analyses were completed, E18.5 embryos were subsequently isolated and phenotyped in situ, without prior fixation.

### 6.2. RNA Isolation and Quantification

Total RNA was isolated from paraffin-embedded hearts, including the ascending aorta, from E18.5 embryos using the Recover AllTM Total Nucleic Acid Isolation Kit (Ambion/RNA by Life technologies, Carlsband, CA, USA), following the manufacturer’s protocol. Briefly, RNA was extracted from 36 embryos that correspond to all possible combinations of VitA-altered diets (VitA-Sup or VitA-Def), embryonic genotypes (Df1/+ or WT), maternal genotypes (Df1/+ or WT), or embryos’ cardiac phenotypes (no malformation or AADs). RNA quantification was performed using a spectrophotometer set at 260 nm. Quality was checked using an Agilent bioanalyzer, and degraded samples were eliminated.

### 6.3. Transcriptome Analysis

For transcriptome analysis, three biological samples were included per group. RNA was analyzed using the Affymetrix Clariom™ (Affymetrix, Inc., Santa Clara, CA, USA) D Assay, a mouse microarray. Microarray hybridization and processing were performed according to the manufacturer’s specifications.

Transcriptome Analysis Console (TAC) Software from Applied Biosystems, which includes LIMMA [66], was used to compare the transcriptomic profiles of embryonic hearts at E18.5, in order to identify differentially expressed genes (DEGs) between VitA-Sup and VitA-Def mouse groups. VitA control groups were also analyzed; however, they were excluded from the differential gene expression comparisons. Briefly, coding genes and miRNAs were filtered, and DEGs of interest were selected based on the following criteria: *p*-value < 0.1 and absolute fold change (FC) greater than 1.2. Non-coding genes were excluded from the subsequent analysis.

#### 6.3.1. Gene Set Enrichment Analysis

Gene Set Enrichment Analysis (GSEA) [67] was performed in order to determine whether Gene Ontology (GO) terms were significantly enriched and whether an entire gene set showed coordinated differences between conditions, rather than focusing on individual genes. Functional enrichment analysis was conducted based on the microarray dataset. The *clusterProfiler* package in R (version 4.4.1) was used to perform the analysis, querying the GO database to identify significantly enriched biological processes (BPs). Enriched terms were filtered using significance thresholds of *p*-value < 0.1 and *q*-value < 0.2. The top 20 most significant BPs were selected by ranking terms according to their absolute Normalized Enrichment Score (NES) in descending order. To reduce redundancy, biological processes belonging to the same parent category were clustered.

#### 6.3.2. Statistical Analysis

A two-way ANOVA was conducted to identify significant differences in serum retinol levels, followed by an LSD test for post hoc analysis when significant (*p* ≤ 0.05). A chi-square test (χ^2^) was used to determine whether a statistically significant relationship existed between categorical variables for which a *p*-value ≤ 0.05 was considered statistically significant. Additionally, multiple testing correction was applied using the Benjamini–Hochberg (BH) procedure to calculate adjusted *p*-values and control the false discovery rate (FDR) for GSEA.

## Figures and Tables

**Figure 1 ijms-26-10595-f001:**
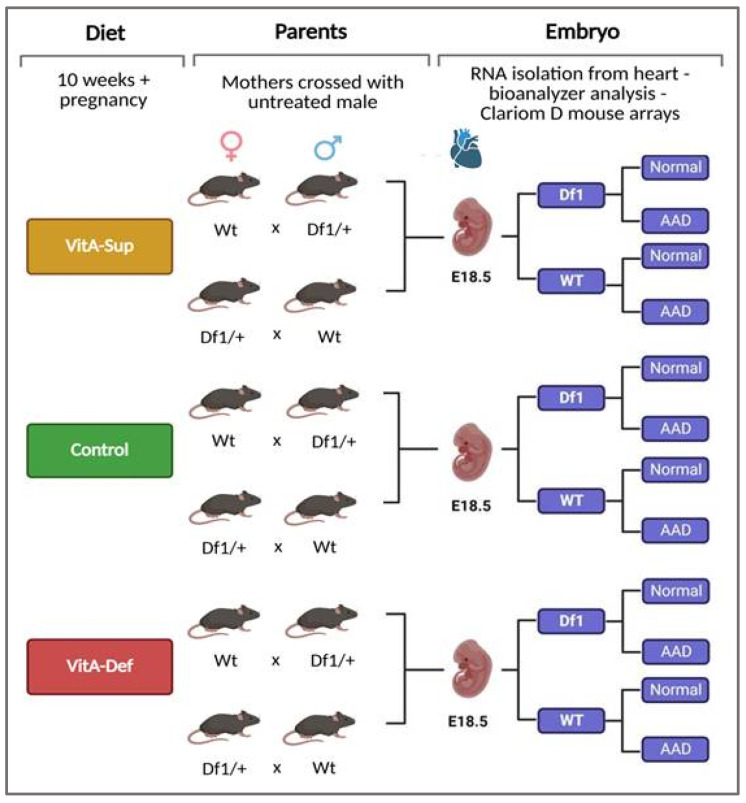
Experimental design and groups. VitA-Sup: vitamin A supplementation in the mother’s diet; Control: standard diet; VitA-Def: vitamin A deficiency in the mother’s diet; E18.5: embryonic day 18.5; WT: wild type; Df1/+: 22q11.2DS mouse model; Normal: no aortic arch defect; AAD: aortic arch defect.

**Figure 2 ijms-26-10595-f002:**
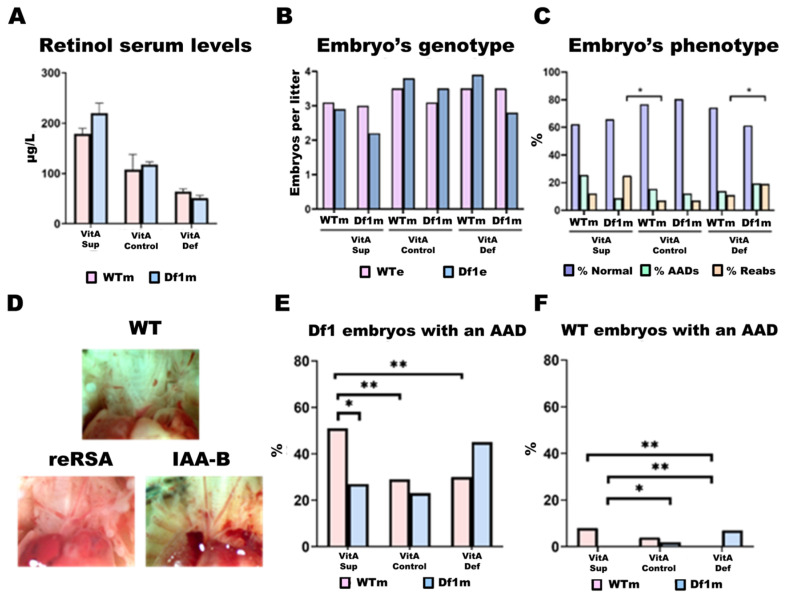
Retinol serum levels and phenotypic results. (**A**) Maternal serum retinol levels (µg/L). A two-way ANOVA was performed to analyze whether differences in retinol serum levels exist between groups. D: diet effect (*p* < 0.05). (**B**) Embryo’s genotype: Number of WT and Df1/+ embryos per litter. (**C**) Embryo’s phenotype: Embryos classified as having either a normal or AAD phenotype and the number of reabsorbed embryos. (**D**) Anatomical structure of the main vessels: Normal phenotype, retroesophageal right subclavian artery (reRSA), and interrupted aortic arch type B (IAA-B). (**E**) Percentage of Df1/+ embryos with AADs per group. (**F**) Percentage of WT embryos with AADs per group. A chi-square test was performed to analyze differences between categorical variables, followed by Fisher’s exact test (**, *p* < 0.05 and *, *p* < 0.1). VitA: vitamin A; sup: supplementation; def: deficiency; WTe: wild-type embryo; Df1e: Df1 embryo’s genotype; WTm: wild-type mother; Df1m: Df1 mother’s genotype; Normal: normal cardiac phenotype; AADs: Aortic arch defects; Reabs: reabsorptions.

**Figure 3 ijms-26-10595-f003:**
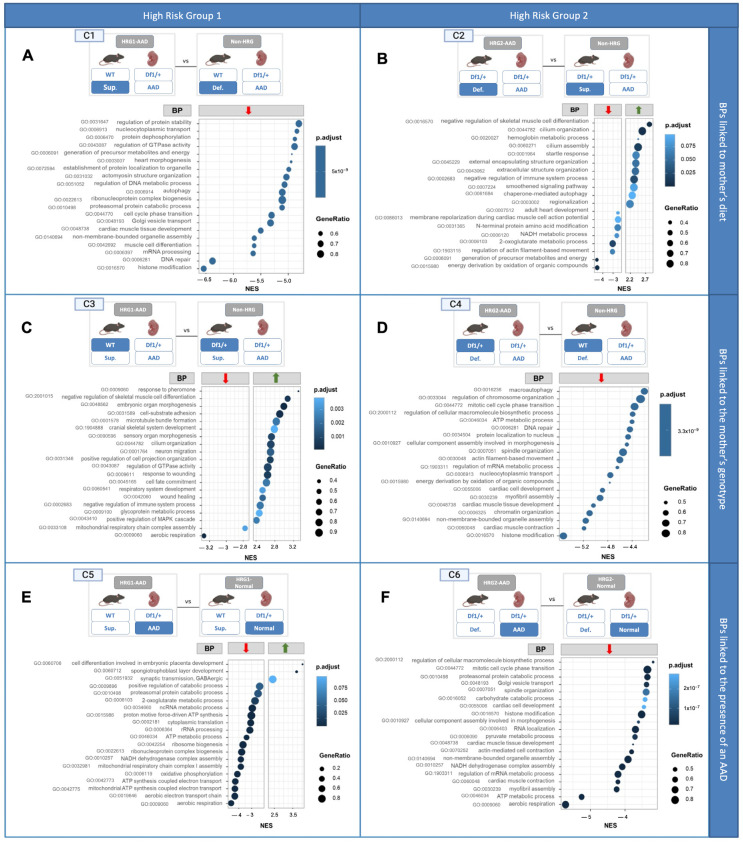
Gene set enrichment analysis when comparing high-risk groups to their pseudo-controls. Subfigures (**A**,**C**,**E**) (**C1**,**C3**,**C5**) graphics show the results of different enrichment analysis for HRG1-AAD and subfigures (**B**,**D**,**F**) (**C2**,**C4**,**C6**) for HRG2-AAD. The main 20 BPs of each comparison are represented according to their NES, gene ratio, and adjusted *p*-value (p-adjust). A lower negative or higher positive NES indicates that downregulation or upregulation of this gene set is more pronounced. Also, redundancy was removed by clustering the processes included within the same parent. Further details on the BPs are provided in Appendix A. HRG1,2-AAD: high-risk group (prevalence of an AAD > 30%); non-HRG1: Df1/+ embryo with an AAD of WT mother fed with VitA-Def; non-HRG2: Df1/+ embryo with an AAD of Df1/+ mother fed with VitA-Sup; BP: biological process; NES: normalized enrichment score; red arrow: BP downregulated; green arrow: BP upregulated.

**Figure 4 ijms-26-10595-f004:**
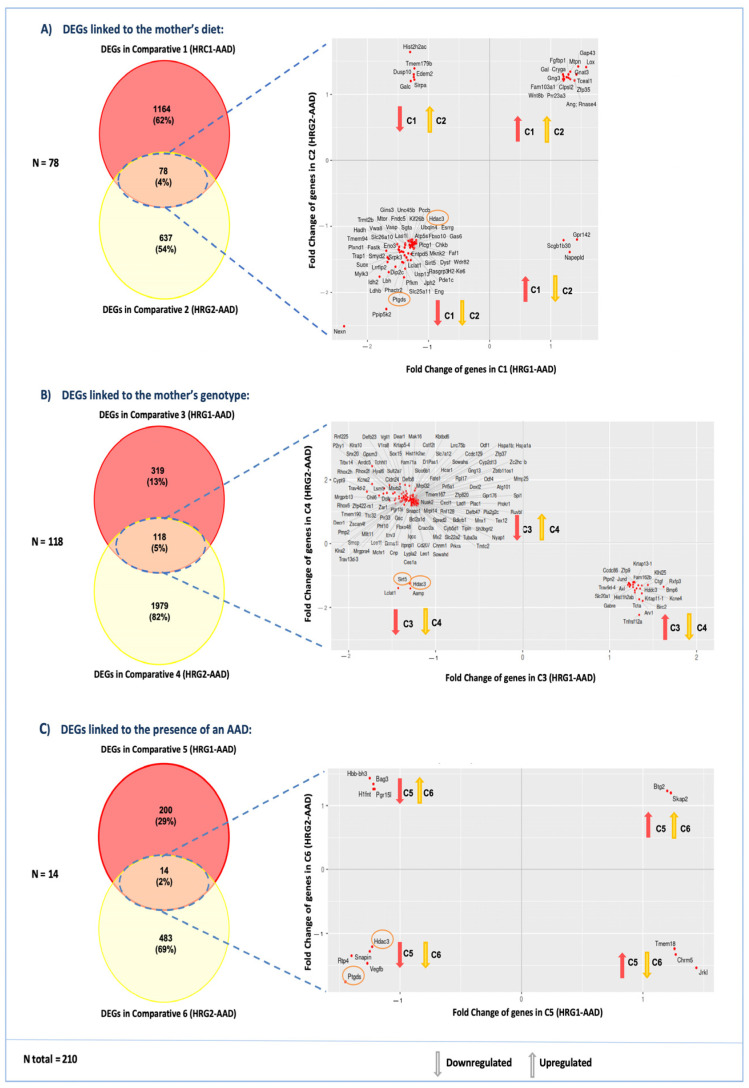
Transcriptomic analysis of HRG1-AAD and HRG2-AAD groups. All genes have been filtered by *p*-value < 0.1 and |fold change| > 1.2. Also, non-coding genes and pseudogenes have been removed. (**A**) Effect of diet: shared DEGs in comparisons 1 and 2, (**B**) effect of maternal genotype: shared DEGs in comparisons 3 and 4, and (**C**) effect of diet–mother genotype interaction on embryo phenotype: shared DEGs in comparisons 5 and 6. In total, 202 DEGs in common without redundancy are the main candidates for causing AADs. The genes circled in yellow are the only ones that are repeated in the three comparative intersections. See Appendix A for more details on the genes and their expression.

**Figure 5 ijms-26-10595-f005:**
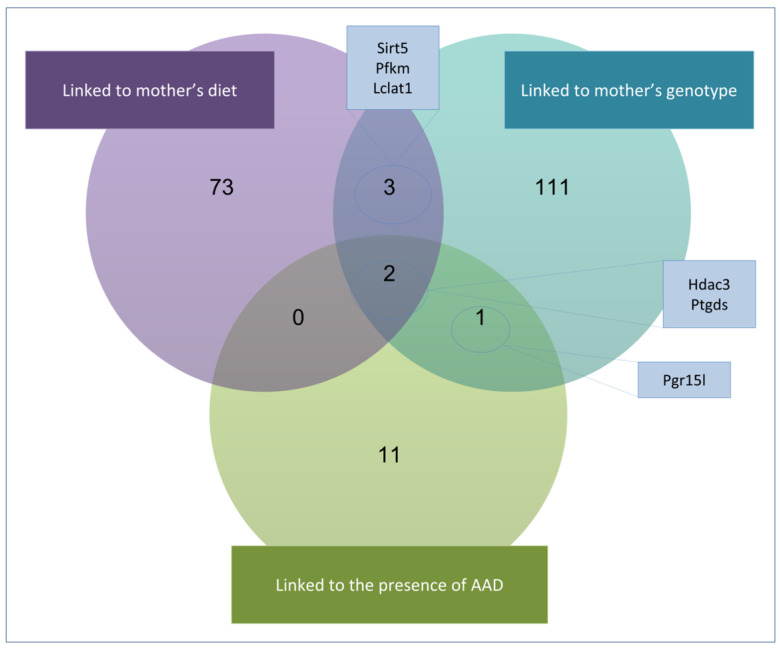
Intersection of shared DEGs between HRG comparisons. A Venn diagram was produced with the shared differentially expressed genes between HRG1 and HRG2 comparisons that were found in the transcriptomic analysis, represented in Figure 3 and Figure 4. This analysis revealed that among the six comparisons displayed in Figure 3, only *Hdac3* and *Ptgds* appeared in all of them (see C1-C6 in Figure 3) suggesting that they may be involved in the pathogenic mechanisms underlying the Df1/+ embryonic phenotype. Both genes were consistently downregulated across all comparisons (see Figure 4).

**Table 1 ijms-26-10595-t001:** Phenotypic data.

Embryo’s Phenotype										
Diet	Mother’s Genotype	Embryo’s Genotype	Total n Embryos	N of AAD	Total n of AAD	N of Normal Phenotype	Total n of Normal	% AAD	% Normal	Litters (n)
VitA Sup.	WT	WT	37	3	21	34	51	8.11	91.89	12
		Df1	35	18		17		51.43	48.57	
	Df1	WT	30	0	6	30	46	0.00	100.00	10
		Df1	22	6		16		27.27	72.73	
VitA Cont.	WT	WT	46	2	16	44	79	4.35	95.65	13
		Df1	49	14		35		28.57	71.43	
	Df1	WT	47	1	13	46	86	2.13	97.87	15
		Df1	52	12		40		23.08	76.92	
VitA Def.	WT	WT	42	0	14	42	75	0.00	100.00	12
		Df1	47	14		33		29.79	70.21	
	Df1	WT	14	1	6	13	19	7.14	92.86	4
		Df1	11	5		6		45.45	54.55	
Total			432	76	76	356	356	17.59	82.41	66

## Data Availability

The raw data generated in this study has been deposited in the ArrayExpress database under ac-cession number E-MTAB-15455.

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
