# Peer review of "Maternal Genotype and Dietary Vitamin A Modify Aortic Arch Phenotypes in a Mouse Model of 22q11DS"

_ijms, 2025, doi:10.3390/ijms262110595_

Round 1

Reviewer 1 Report

Comments and Suggestions for Authors

Amengual-Cladera et al. submitted the paper : Maternal Genotype and Dietary Vitamin A Modify Aortic Arch 2 Phenotypes in a Mouse Model of 22q11DS 3

Minor comments:

Abstract: ok clear, well written

-Introduction, line 61; correct, but as many different non numerical chromosomal changes can cause HD, you may consider modifying  “like Down syndrome (trisomy 21), Turner syndrome (monosomy X), many other structural chromosomal anomalies and pathogenic….”

-You may also consider quoting possible monogenic causes oh HD, to remember that in rare instances monogenic HD are observed.

Prendiville, T., Jay, P. Y., & Pu, W. T. (2014). Insights into the genetic structure of congenital heart disease from human and murine studies on monogenic disorders. Cold Spring Harbor perspectives in medicine4(10), a013946. https://doi.org/10.1101/cshperspect.a013946

-Overall introduction OK

Results 2.1

I am interested o a comment about the anatomy of the AAD; is it easy to define clearly presence /or absence of AAD? There no minor defects difficult to classify?

2.4

-Line 230: “conversely” pleaae check if it is the best word to express your idea.

Discussion

-Line 465: please consider modify as  “deficiency was observed”

Author Response

Comment 1: Abstract: ok clear, well written  

Answer 1: We appreciate the reviewer’s positive feedback on the abstract. 

Comment 2: Introduction, line 61; correct, but as many different non numerical chromosomal changes can cause HD, you may consider modifying “like Down syndrome (trisomy 21), Turner syndrome (monosomy X), many other structural chromosomal anomalies and pathogenic….”  

 -You may also consider quoting possible monogenic causes of HD, to remember that in rare instances monogenic HD are observed.  

 Prendiville, T., Jay, P. Y., & Pu, W. T. (2014). Insights into the genetic structure of congenital heart disease from human and murine studies on monogenic disorders. Cold Spring Harbor perspectives in medicine, 4(10), a013946. https://doi.org/10.1101/cshperspect.a013946 

Answer 2: We acknowledge the reviewer’s comment and we have considered this suggestion in our revision. We have introduced this to address the reviewer’s comment: “Most of non-syndromic CHD occurs sporadically, with a very scarce number of families having a clear monogenic inheritance. This makes the identification of the genetic cause very difficult using a classical genetic approach.” (line 60 in the new version) 

Comment 3: Results 2.1: I am interested o a comment about the anatomy of the AAD; is it easy to define clearly presence /or absence of AAD? Are there no minor defects difficult to classify? 

Answer 3: We appreciate the reviewer’s observation. The most frequent aortic arch defects observed in our experimental model are the development of aberrant right subclavian arteries (re-RSA) (n= 56 embryos), interrupted aortic arches type B (IAA-B) (n= 17 embryos), occasional right aortic arches, and, in some cases, the concomitant presence of both IAA-B and re-RSA (n= 4 embryos). However, the macroscopic phenotyping of type A or C interrupted arches or aortic coarctation would be feasible at E18.5, the developmental stage at which phenotyping was performed in our study. For this reason, we are confident that all existing alterations related to the most frequent aortic arch defects were adequately recorded. In addition, histological analyses were carried out to rule out morphological alterations in the internal cardiac structures (i.e., non-AAD) in a subset of embryos.  Furthermore, this animal model has been extensively studied and the main or solely described cardiac phenotypes are aortic arch anomalies (see references in manuscript by Dr. Baldini’s group, for example Lindsay and Baldini, 2001).

  1. A. Lindsay et al., “Tbx1 haploinsufficiency in the DiGeorge syndrome region causes aortic arch defects in mice,” Nature, vol. 410, no. 6824, pp. 97–101, Mar. 2001, doi: 10.1038/35065105;KWRD=SCIENCE.

Comment 4: 2.4: -Line 230: “conversely” please check if it is the best word to express your idea. 

Answer 4: Thank you for pointing this out. We agree with this comment. In this paragraph, the use of ‘conversely’ may be misleading, as the statement does not present a contrasting result but rather an additional observation. We have considered using ‘In addition’ instead, as it seems more appropriate in this context (line 236). 

Comment 5: Discussion: -Line 465: please consider modify as “deficiency was observed”. 

Answer 5: Thank you for highlighting this point. We have changed it to make the sentence clearer in line 470 of the new version: “Vitamin A supplementation was associated with a higher incidence of AADs only in embryos from WT mothers, whereas under deficiency, increased AAD numbers were observed exclusively in embryos from Df1/+ mothers (Fig. 2E).” (line 469 of the new version)

Reviewer 2 Report

Comments and Suggestions for Authors

The authors presented the results of experimental studies on genotype-environment interactions in 22q11.2DS. Their data suggest that both maternal diet and genotype may influence the phenotype of mouse embryos with a deletion corresponding to DiGeorge syndrome. The results expand on earlier data on embryo genotype. Findings suggest that maternal factors can modulate the severity of CHD

Minor comments

  1. Transcriptome analysis only at stage E18.5 – it is unknown whether the changes are causal or secondary. Limitations, in fact, are provided.
  2. No histopathological confirmation of phenotypes (e.g., 3D cardiac imaging or microtomography). The authors describe visual assessment of the phenotype in most embryos and histology in only 55 cases. No microtomography or quantitative methods were used, which limits the reliability of phenotyping. This should be included in Limitations.
  3. It is unclear whether the effects are specific to AAD or also apply to other heart defects.
  4. It would be worthwhile to include serum and tissue retinol/retinoid levels to confirm the biological effect of the diet.
  5. Lines 646- 649 include parts from template and should be omitted.

Author Response

Comment 1: Transcriptome analysis only at stage E18.5 – it is unknown whether the changes are causal or secondary. Limitations, in fact, are provided. 

Answer 1: As the reviewer stated, this has been acknowledged in the limitations section at the end of the manuscript (section 5. Limitations of research, line 567-574). 

Comment 2: No histopathological confirmation of phenotypes (e.g., 3D cardiac imaging or microtomography). The authors describe visual assessment of the phenotype in most embryos and histology in only 55 cases. No microtomography or quantitative methods were used, which limits the reliability of phenotyping. This should be included in Limitations. 

Answer 2: We acknowledge the reviewer’s comment. However, at embryonic day 18.5 (E18.5), the aortic arch and its main branches are already fully developed and of sufficient size to allow reliable macroscopic phenotyping under a stereomicroscope. At this stage, aberrant positioning or absence of major vessels is easily identifiable, making visual assessment a well-established and widely used approach for detecting aortic arch defects (AADs).  In addition, there is extensive literature that shows that the main and solely described cardiac phenotypes of Df1/+ embryos are aortic arch anomalies (see references in manuscript by Dr. Baldini’s group, for example Lindsay and Baldini, 2001).

  1. A. Lindsay et al., “Tbx1 haploinsufficiency in the DiGeorge syndrome region causes aortic arch defects in mice,” Nature, vol. 410, no. 6824, pp. 97–101, Mar. 2001, doi: 10.1038/35065105;KWRD=SCIENCE.

In addition, histological analyses using hematoxylin and eosin (H&E) staining were performed in 55 embryos with the specific aim of ruling out potential structural cardiac alterations beyond AADs. These analyses were limited to this subset because the purpose was solely to confirm the absence of additional defects, which, moreover, have never been reported in this experimental model.  

We agree, however, that the absence of complementary 3D imaging or microtomography may represent a limitation, and we have included this consideration in the Limitations section of the manuscript. 

Comment 3: It is unclear whether the effects are specific to AAD or also apply to other heart defects. 

Answer 3: Thank you for bringing this to our attention. In our histological examinations no septal or other internal structural heart abnormalities were found, we only observed AAD such as reRSA/ARSA, (n= 56 embryos), IAA-B (n= 17 embryos), the combination of ARSA and IAA-B (n= 3 embryos) and the combination of ARSA and a right aortic arch (n=1 embryo) (Fig. 2D). Based on these results, we considered that the effects are linked to AAD but not to other heart defects. Nevertheless, for future studies, it would be worth considering an increased sample size and the inclusion of additional phenotyping techniques to improve visual assessment. 

Comment 4: It would be worthwhile to include serum and tissue retinol/retinoid levels to confirm the biological effect of the diet. 

Answer 4: We appreciate the reviewer’s observation. However, the results of the maternal serum VitA analysis are already presented in section ‘2.1.1. Vitamin A in the diet determines retinol leves in blood’ and the details of the analysis were included in the Material and methods, section 6.1. Mice in diets. Also, these results were discussed in section 3. Discussion, lines 477-480. 

Comment 5: Lines 646- 649 include parts from template and should be omitted. 

Answer 5: Thank you for bringing this to our attention, we missed this in our revision and have now deleted it.